# More Details, Please: Improving Autoformalization with More Detailed Proofs

Guillem Tarrach [1]   Albert Q. Jiang [1]   Daniel Raggi [1]   Wenda Li [2]   Mateja Jamnik [1]

## Abstract

The formalization of mathematical theorems and their proofs is a time-consuming and tedious process which, despite recent advances in the reasoning capabilities of AI systems, remains a challenging task for computers. Existing attempts to automate the process with language models struggle with the difference in level of detail between formal and informal proofs. Successful autoformalization requires models to understand and be able to explain the nuances of logical arguments, a critical aspect of reasoning that is often overlooked in existing research. In this work, we introduce *Sketch, Prove, Add Details & Repeat* (SPADER), an approach that enhances proof autoformalizers by using language models to infer and explicitly incorporate implicit details from informal proofs. With the same number of autoformalization attempts, our method increases the percentage of successfully formalized problems in the miniF2F test dataset from 34.8% to 38.1%.

## 1. Introduction

A significant body of recent work has investigated the reasoning capabilities of Large Language Models (LLMs), particularly in the context of solving mathematical problems. One frequently studied task is Automated Theorem Proving (ATP), which involves automatically generating formal proofs of mathematical theorems. However, few studies have investigated the ability of LLMs to understand and explain mathematical arguments. In this work, we introduce an approach that leverages this capability to construct more detailed informal mathematical proofs, thereby improving the process of autoformalization – the translation of informal proofs into formally verifiable formal proofs. Informal proofs lack many details that are necessary to verify their correctness. While formal proofs do not suffer from this issue, in practice the focus on low-level details makes formal automated theorem provers less successful at high-level planning. As a result, autoformalization systems struggle with the discrepancy in the level of detail in formal and informal proofs (Jiang et al., 2023, Section 5.2 and Appendix C). Our approach uses LLMs to explain informal proofs by inferring and incorporating implicit details, thereby bridging the gap between informal and formal proofs.

To plan ahead and focus on the overall proof strategy, mathematicians usually write proofs in a non-linear, hierarchical manner: They start by writing a high-level proof draft and iteratively add more detail until the proof is considered complete. Previous work on language model-based ATP has studied such hierarchical set-ups (Li et al., 2021; Jiang et al., 2023; Mikuła et al., 2023), but has not explored adding detail to informal proofs. For example, in *Draft, Sketch and Prove* (DSP) (Jiang et al., 2023), a high-level informal proof draft is used to inform a more detailed formal proof sketch, which is later completed by an automated theorem prover. A common error case occurs in the process of translating informal drafts into formal sketches. This process happens in a single step: the model must decide which steps in the draft need further argumentation, add the missing details, and translate the informal draft to the formal language all at once. Therefore, the approach could benefit from using specialized models for each of the three stages, particularly adding the missing details as it is especially complex.

Our main contribution is SPADER, a method that enhances autoformalizers through the use of LLMs to construct more detailed informal mathematical proofs by incorporating the implicit reasoning steps into them. The approach is illustrated in Figure 1. Starting with a theorem statement and an informal proof, we use an autoformalizer to generate a formal proof sketch. We attempt to complete the sketch using an automated prover. If some steps cannot be proved, we use an LLM to provide more details about them and re-attempt the process with a new, more detailed, informal proof. The process succeeds if the additional detail is correct and offers a good explanation for the problematic step.

Our experiments show that SPADER increases the success rate of autoformalization systems. With the same number of autoformalization attempts, adding detail to informal

---

[1]University of Cambridge [2]University of Edinburgh. Correspondence to: Guillem Tarrach <guillem.tarrach@gmail.com>.

*AI for MATH Workshop at ICML 2024*, Vienna, Austria. Copyright 2024 by the author(s).

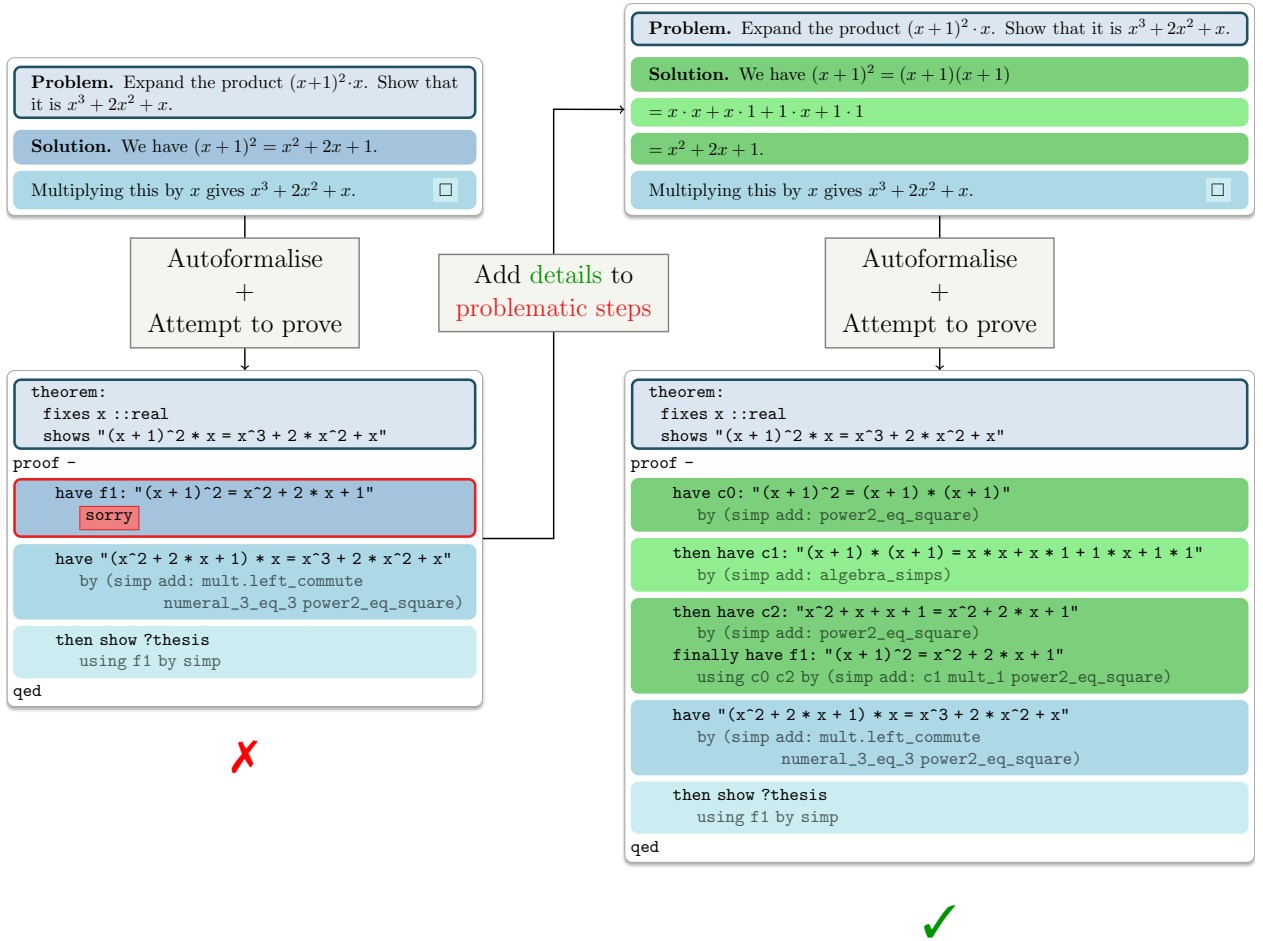

*Figure 1.* **Overview of SPADER (*Sketch, Prove, Add Details & Repeat*)**. Given formal and informal statements of a theorem and an informal proof, we attempt to autoformalize the proof and then formally verify it. Whenever a particular step cannot be proved, we add more details to it using a language model. We re-attempt the formal verification with the more detailed informal proof. The inclusion of more details into informal proofs improves the performance of the autoformalizer.

proofs with GPT-4o increases the number of successfully verified problems in the miniF2F test from 85 (34.8%) to 93 (38.1%).

In conclusion, we make the following contributions:

- We propose a method for using LLMs to construct more detailed proofs by inferring implicit details in informal mathematical proofs.

- We demonstrate the usefulness of the presented method for autoformalization.

Our work shows that LLMs can provide detailed mathematical proofs by inferring and explaining implicit reasoning steps. This ability helps bridge the gap between informal and formal mathematical proofs and enables LLM-based autoformalization systems to verify more theorems.

## 2. Background and Related Work

### 2.1. Mathematical Reasoning with Language Models

With recent advances in language models, particularly the introduction of LLMs, there has been an increase in research into their reasoning capabilities, particularly in the context of mathematical problem-solving (Hendrycks et al., 2021; Drori et al., 2022; Welleck et al., 2021). While alternative prompting methods (Wei et al., 2022; Yao et al., 2023; Zheng et al., 2023a) help improve the accuracy of reasoning arguments, language models still frequently make mistakes. These challenges highlight the need for robust verification methods to complement informal reasoning.

Furthermore, the ability of LLMs to understand and explain existing arguments remains largely unexplored. In this work, we investigate these abilities and their evaluation

through autoformalization and formal verification.

## 2.2. Autoformalization

To address the limitations of reasoning with language models, recent work has explored the combination of informal reasoning with formal verification through autoformalization. While early approaches to autoformalization with deep learning took inspiration from Neural Machine Translation (Wang et al., 2018), it has been observed (Wu et al., 2022) that LLMs are better suited for this task because of their in-context few-shot learning capabilities (Brown et al., 2020) and the scarcity of parallel informal-formal data. *Draft, Sketch and Prove* (DSP) (Jiang et al., 2023) approaches automated theorem proving by autoformalizing computer-generated informal proofs. Autoformalization proceeds in two stages. The first stage uses an LLM to generate a formal sketch that follows the informal proof. The formal sketch is not a complete formal proof; instead, it outlines the overall proof strategy by describing intermediate conjectures. In the second stage, an off-the-shelf formal theorem prover is employed to prove the intermediate conjectures in the sketch, thus completing the proof. However, in many cases, the informal proof does not contain enough detail for the automated prover to fill in the gaps. *Don't Trust: Verify* (Zhou et al., 2024) applies a similar method to open-ended mathematical problems. The method consists of generating an informal chain-of-thought reasoning argument to find the answer to a problem, which is considered valid only if it can be autoformalized and formally verified by an automated prover. This approach suffers from the same problem, where informal solutions sometimes fail to be verified despite being correct. *Lyra* (Zheng et al., 2023b) addresses this issue by prompting LLMs to repair errors in the formal proofs. In our approach, we instead prompt LLMs to add more detail to the informal proofs. We note that informal solutions to the test problems used to evaluate the methods in Lyra, as well as ours, may be part of the training data for the LLMs used (GPT-4 and GPT-4o. respectively). Therefore, these methods are better understood as methods for autoformalization rather than theorem-proving.

## 2.3. Formal Theorem Proving

The construction of mathematical proofs in non-linear ways, where detail is dynamically added to problematic steps, has been more widely studied in the context of formal theorem proving. *IsarStep* (Li et al., 2021) introduces a benchmark for the task of generating intermediate steps in a formal proof. *Magnushammer* (Mikuła et al., 2023) combines a premise selection model with formal proof generators (Jiang et al., 2022). The premise selection model is employed to find premises that imply the intermediate conjectures generated by the proof generator, which together constitute a proof of the theorem. *Baldur* (First et al., 2023) approaches

ATP by generating a full formal proof with a pre-trained language model and using a specialized model to repair any errors with it. However, unlike the methods in Section 2.2, none of these methods make use of informal mathematical data, which is significantly more abundant than formal data.

# 3. SPADER: Enhancing Autoformalization with More Detailed Informal Proofs

We now describe SPADER (*Sketch, Prove, Add Details & Repeat*), our approach to autoformalization and formal verification. This approach enhances the performance of autoformalizers by using LLMs to construct more detailed proofs that guide the autoformalization and formal verification process. The approach is illustrated in Figure 1 and summarized in Algorithm 1. We assume the user has access to an autoformalizer and an automated theorem prover or proof assistant. Given an informal proof – a *proof draft* in the terminology of DSP (Jiang et al., 2023) – the approach consists of the following stages:

**Stage 1 (Sketch).** The informal proof draft is translated into a formal sketch using an autoformalizer. The formal sketch need not be a complete formal proof; it may contain open conjectures that will be handled in the next stage. The formal sketch should follow the high-level structure of the informal draft. For example, in Figure 1, the different intermediate conjectures in the formal proof can be mapped to steps with the same color in the informal draft. This may be achieved through the use of comments in the sketch.

**Stage 2 (Prove).** An automated theorem prover attempts to fill in the missing details in the formal sketch, thus completing the proof. If a proof is found, the process has been successful. If the theorem prover is unable to prove an intermediate step, the step is flagged as not proven, and the theorem prover proceeds with the rest of the proof, assuming that the step is true.

**Stage 3. (Add Detail).** The steps in the formal sketch that could not be proved are mapped to the corresponding steps in the informal draft. A separate model, typically an LLM, is then prompted to provide more details on the steps in question and to generate a more detailed informal draft.

**Repeat.** The process is repeated starting from Stage 1 with the new draft.

In Figure 1, the vertical downward arrows represent Stages 1 and 2, and the middle arrow represents Stage 3. In the rest of this paper, we refer to the number of times that Stage 3 is run as the number of *detailing passes $M$*.

**Algorithm 1** SPADER (Sketch, Prove, Add Details & Repeat). The algorithm assumes that the user has access to an autoformalizer `autoformalize`, an automated theorem prover `attempt_formal_proof`, and a model `add_detail` that can add detail to proofs.

---

**Parameters:** Number of detailing passes $M$.
**Input:** Theorem $\mathbf{t}$, informal proof $\mathbf{p}$.
$draft \leftarrow \mathbf{p}$
**for** $j \in \{0, \ldots, M\}$ **do**
  $sketch \leftarrow$ `autoformalize`$(\mathbf{t}, draft)$
  $proof \leftarrow$ `attempt_formal_proof`$(sketch)$
  $failedSteps \leftarrow \{s \in proof : s.proven = false\}$
  **if** $failedSteps = \emptyset$ **then**
    **return** $proof$
  **else if** $j < M$ **then**
    $draft \leftarrow$ `add_detail`$(\mathbf{t}, draft, failedSteps)$
  **end if**
**end for**
**return** FAIL

---

## 4. Experiments

Next, we describe the experiments we conducted to evaluate whether adding detail to informal proofs with SPADER improves the performance of autoformalizers.

### 4.1. Dataset and Metrics

We evaluate SPADER on the miniF2F dataset (Zheng et al., 2021). The miniF2F dataset comprises 488 mathematical problems (244 validation problems and 244 test problems). Each problem consists of a collection of formal statements in different formal languages: Isabelle (Paulson, 1988), Lean (de Moura et al., 2015), MetaMath (Yu et al., 2024) and HOL Light (Bansal et al., 2019). This dataset was expanded in DSP (Jiang et al., 2023) to include an informal statement and a human-written informal solution for each formal statement. Our goal is to correctly autoformalize and formally verify the informal solution in Isabelle. We evaluate the performance of our method according to the number of test problem solutions that can be correctly formalized and verified.

### 4.2. Implementation

We performed our experiments with the Isabelle proof assistant (Paulson, 1988). We have considered $M = 1$ and $M = 2$ detailing passes. This allows us to compare the effect of multiple detailing passes.

As our autoformalization model, we use GPT-4o[1]. We prompt the model through the OpenAI API to translate the informal proof into a formal Isabelle/HOL sketch with

---

[1] https://openai.com/index/hello-gpt-4o/

3-shot prompting. We prompt the model to include the original proof as comments before the corresponding steps in Isabelle. We also prompt the model to include a comment concluding the informal proof (with "The result follows") so that the end of the proof can be marked as needing more detail in the next stages. We include 3 in-context examples, randomly sampled from a list of 17 hand-labeled samples, which are modified versions of those from (Jiang et al., 2023). They have been modified to break down the informal proofs (included as comments in the formal proof) into smaller steps. We hope that segmenting informal proofs into smaller steps makes it easier to pinpoint the problematic steps later. To generate a diverse set of sketches across multiple runs, we generate them with temperature sampling with a temperature parameter of 0.6 as in DSP (Jiang et al., 2023).

We attempt to complete the sketch using the Isabelle theorem prover. Whenever Isabelle fails to prove an intermediate conjecture, we attempt to prove it with several heuristics, as in DSP (Jiang et al., 2023), and Sledgehammer (Paulson & Blanchette), a collection of automated theorem provers. If these fail to prove a conjecture, we add a `sorry` statement, which tells Isabelle to assume that the step has been proven and allows it to continue verifying the rest of the proof. If the verification process encounters an error (e.g. if the formal sketch contains syntax errors), we abort and stop the run. If a proof is parsed correctly but contains `sorry` statements, we add an 'unproven' flag to the last comment before each such statement. Since the comments contain the original informal proof, the flagged steps are those that require more detail. We concatenate all the steps to recover the original proof and surround the flagged steps with the strings `<MORE_DETAIL>` and `</MORE_DETAIL>`.

To add detail to the proof, we prompt GPT-4o to rewrite the proof with more detail in the marked steps. We use temperature sampling with a temperature parameter of 0.4. We have used a lower temperature parameter than before to prioritize accuracy over diversity since only one detailed draft is generated per sketch. We generate either one (if $M = 2$) or two (if $M = 1$) formal sketches for each of the new drafts and re-attempt the formal verification. If $M = 2$, we repeat the process of adding detail to unproven steps, generating a formal sketch, and attempting to complete it.

We ensure that the number of autoformalization attempts is consistent across the different experiments ($M = 1$, $M = 2$ and the baseline) by implementing a modified version of Algorithm 1, which is described in the appendix.

We have run each experiment $N = 100$ times and consider a problem solution to be successfully autoformalized whenever one of these succeeds. To avoid confusion in our discussion, we will distinguish between *individual runs* (each run of an experiment) and *autoformalization attempts* (each time the autoformalizer is called). Each individual run

*Table 1.* **Results of the experiments on autoformalization.** The table shows the number of problems correctly formalized in the miniF2F test dataset for different autoformalization methods. With the same number of autoformalization attempts, SPADER is able to write complete formal solutions for 8 more problems than the Sketch and Prove baseline, where additional details are not added to informal drafts.

| METHOD | PROBLEMS FORMALIZED |
|---|---|
| SKETCH AND PROVE | 85 (34.8%) |
| SPADER, $M = 1$ (OURS) | **93 (38.1%)** |
| SPADER, $M = 2$ (OURS) | **93 (38.1%)** |

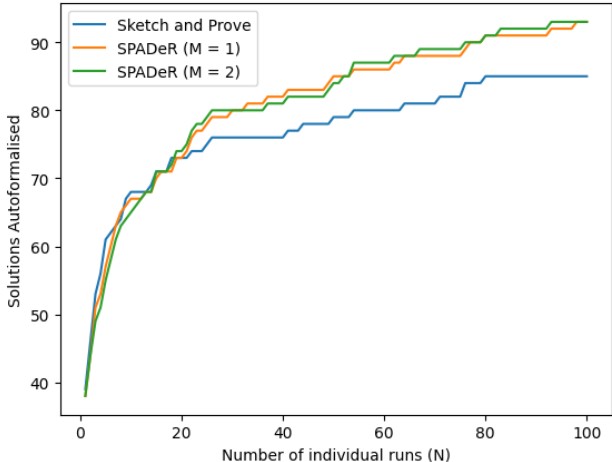

*Figure 2.* **Number of problems solved in the miniF2F test set for different numbers of runs** for SPADER with $M = 2$ (green), $M = 1$ (orange), and the Sketch and Prove baseline (blue). We use the same number of autoformalization attempts in all methods. After the easier problems are solved, SPADER can autoformalize more problems.

may generate up to three sketches with different levels of detail, so the process involves at most 300 autoformalization attempts. Since we stop a run whenever a formal proof is found or the verification encounters an error, this bound is rarely reached in practice (with the average number of attempts being 108).

### 4.3. Baselines

As a baseline, we compare our approach against our implementation of the autoformalization method presented in DSP (Jiang et al., 2023): we first generate a formal sketch, which we try to formally verify in Isabelle using the same procedure described above. We refer to this baseline as *Sketch and Prove* since it does not include the *Draft* stage in DSP: we conduct our experiments exclusively with human-generated proofs. These proofs, unlike computer-generated ones, are known to be correct and contain no errors. This allows us to isolate the effect of additional detail on autoformalization more accurately. As discussed above, for each individual run, we use the same number of autoformalization attempts in the baseline and the other approaches.

### 4.4. Results

The results of our experiments on autoformalization are displayed in Table 1. With the same number of autoformalization attempts, SPADER achieves a higher success rate than the Sketch and Prove baseline. We note that SPADER is able to solve the same number of problems with $M = 1$ and $M = 2$, which suggests that adding even more detail to detailed sketches does not improve performance. Figure 2 shows how the number of successfully solved problems changes with the number of individual runs $N$.

## 5. Discussion

**LLMs Can Understand and Explain Informal Proofs.** In our experiments, we have asked LLMs to provide more details on specific steps in mathematical proofs. For 8 prob-

lems in the miniF2F dataset, the new, more detailed proofs could be formalized, while the original proofs could not. Therefore, the details added by the LLM must be correct (since they have been formally verified) and must provide a good explanation of the arguments in the original proof (since they help verify the rest of the proof). We have included a few successful examples in the appendix.

**Additional Details do not Improve Autoformalization on Easy Problems.** We observe from Figure 2 that for small numbers of runs ($N < 20$), the more detailed proofs do not improve the success rate of autoformalizers. All the problems that are proved by the baseline, but not by SPADER in this range, are solved in subsequent runs by SPADER on initial autoformalization attempts (i.e., by autoformalizing the original drafts, which do not contain additional details). This suggests that the process of adding detail is helpful only for difficult problems. Future research may explore distinguishing which proofs benefit most from additional detail so that resources can be allocated more effectively.

**Multiple Detailing Passes are not Necessary.** Table 1 and Figure 2 indicate that adding detail twice ($M = 2$) does not yield better performance than adding detail once ($M = 1$). As discussed above, the performance of the baseline (which corresponds to $M = 0$) is similar to that of $M = 1, 2$ for small numbers of runs, suggesting that the similar trend for $M = 1$ and $M = 2$ might not hold for a very large number of runs ($N \gg 100$). However, verifying or making use of this would require an impractical amount of computational resources. It is also possible that more detailing passes are beneficial for more complicated informal proofs, since they

are usually less detailed.

**Use of Specialized Models for Adding Detail.** We have experimented with using specialized models that retrieve relevant references or insert intermediate steps in proofs to provide more detail. However, these models have not proved successful at improving autoformalization. We believe this is due to the difference in distribution between the problems in the miniF2F dataset, consisting of high-school level mathematics and employing rich language in their solutions, and our training data, which is collected from ProofWiki[2] and consists of university-level mathematics using more rigid language.

## 6. Conclusion

To create successful autoformalization systems, it is essential to reconcile the lack of detail of informal proofs with the high detail requirements of formal verification systems. In this paper, we introduced SPADER, an approach that enhances autoformalizers by using Large Language Models (LLMs) to construct more detailed informal mathematical proofs. By inferring and incorporating implicit details in proofs, this approach improves the accuracy of language model-based autoformalizers. This shows that LLMs possess the ability to understand and explain existing mathematical arguments.

## Impact Statement

This paper presents work whose goal is to advance the field of Machine Learning. There are many potential societal consequences of our work, none of which we feel must be specifically highlighted here.

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

# A. Implementation Details

## A.1. Achieving a Consistent Number of Autoformalization Attempts

To ensure that the baseline is on a level playing field with our method, we have modified Algorithm 1 to use the same number of autoformalization attempts for the baseline and our method. The resulting modification can be found in Algorithm 2. We use $M^* = 2$, so that $M = 1, 2$ correspond to our implementation of SPADER and $M = 0$ corresponds to the baseline with the same number of attempts.

---

**Algorithm 2** SPADER with a consistent number of attempts for different detailing passes $M$. The algorithm assumes that the user has access to an autoformalizer `autoformalize`, an automated theorem prover `attempt_formal_proof`, and a model `add_detail` that can add detail to proofs.

---

**Parameters:** Number of maximum detailing passes $M^*$.
**Input:** Theorem $\mathbf{t}$, informal proof $\mathbf{p}$.

**for** $M \in \{0, \ldots, M^*\}$ **do**
  $successfulProofs[M] \leftarrow \emptyset$
**end for**
$drafts[1] \leftarrow \mathbf{p}$
/* Autoformalize with $M = M^*$ detailing passes */
**for** $j \in \{0, \ldots, M^*\}$ **do**
  $sketch \leftarrow$ `autoformalize`$(\mathbf{t}, drafts[j])$
  $proof \leftarrow$ `attempt_formal_proof`$(sketch)$
  **if** $proof = $ ERROR **then**
    **break**
  **end if**
  $failedSteps \leftarrow \{s \in proof : s.proven = false\}$
  **if** $failedSteps = \emptyset$ **then**
    $successfulProofs[M^*]$.add$(proof)$
    **break**
  **else if** $j < M^*$ **then**
    $drafts[j+1] \leftarrow$ `add_detail`$(\mathbf{t}, drafts[j], failedSteps)$
  **end if**
**end for**
/* Autoformalize with $M < M^*$ detailing passes with the same number of autoformalization attempts */
**for** $M \in \{0, \ldots, M^* - 1\}$ **do**
  **for** $j \in \{0, \ldots, \text{length}(drafts)\}$ **do**
    $sketch \leftarrow$ `autoformalize`$(\mathbf{t}, drafts[\min(M, j)])$
    $proof \leftarrow$ `attempt_formal_proof`$(sketch)$
    $failedSteps \leftarrow \{s \in proof : s.proven = false\}$
    **if** $failedSteps = \emptyset$ **then**
      $successfulProofs[M]$.add$(proof)$
      **break**
    **end if**
  **end for**
**end for**
**return** FAIL

---

## A.2. Autoformalization with LLMs

We describe in detail our process for translating informal proofs into formal sketches with GPT-4o. In the initial sketching stage ($j = 0$ in Algorithm 2), where we work with the original informal proofs (as opposed to more detailed ones), we have used two different prompts that ask the model to write a formal sketch in Isabelle that follows the informal draft, where each informal step is included as a comment before the corresponding formal steps. The first has very detailed instructions,

prompting the model to follow the informal proof very closely and make the informal steps (included in comments) as close as possible. The motivation behind this is that, with smaller steps, our approach will be able to more accurately zone in on the parts of the proof that are difficult for the automated prover. In contrast to (Jiang et al., 2023), instead of prompting the model to invoke the Sledgehammer automated prover to prove intermediate conjectures whenever possible, we allow the model to predict premises that will prove it. The second prompt contains less detailed instructions.

We sample 3 random examples from a list of 17 hand-labeled samples and include them as in-context examples. The examples contain the original informal proof as comments: for each step in the informal proof, a comment containing it is followed by the corresponding formal statement. The examples are based on the examples from (Jiang et al., 2023); however, we segment the informal proofs into smaller steps. For each set of examples and prompts, we have generated two outputs with temperature sampling with a temperature parameter of 0.6 and a maximum output context length of 1024 tokens. We did not observe any significant difference in the performance of the two prompts in our validation runs. For the following sketching stages ($j > 0$), we use only the first prompt and generate a single output per set of in-context examples, also with a temperature parameter of 0.6 and a maximum output context length of 1024 tokens.

### A.3. ATP heuristics

Whenever an intermediate conjecture in the formal sketch fails to be proved, we attempt to prove it with the following heuristics: `auto`, `simp`, `blast`, `fastforce`, `force`, `eval`, `presburger`, `sos`, `arith`, `linarith`, `auto simp: field_simps algebra_simps`. Note that, differently to DSP (Jiang et al., 2023), we include `algebra_simps` in the last one. If the heuristics fail, we attempt to prove the step with the Sledgehammer automated prover (Paulson & Blanchette). We interact with Isabelle from Python scripts via Portal to ISAbelle (Jiang et al., 2021).

## B. Examples of Successfully Autoformalized Proofs

Below, we include four examples of problems whose original human-written solution could not be autoformalized with the Sketch and Prove baseline, but which could be autoformalized with the additional detail provided by GPT-4o with SPADeR. We include the original informal statement and original human-written informal solution, as well as the more detailed informal solution and complete formal solutions provided by SPADeR. We show two number theory problems (Figure 3 and Figure 4) and two algebra problems (Figure 5 and Figure 6). The first problem (Figure 3) was discussed in Appendix C of DSP (Jiang et al., 2023) as an example of a problem whose human-written solution could not be correctly autoformalized due to not containing enough detail, but for which a computer-written solution could.

**Problem.** Prove that the fraction $\frac{21n+4}{14n+3}$ is irreducible for every natural number $n$.

**Original informal solution.** Denoting the greatest common divisor of $a, b$ as $(a, b)$, we use the Euclidean algorithm:
$(21n + 4, 14n + 3) = (7n + 1, 14n + 3) = (7n + 1, 1) = 1$
It follows that $\frac{21n+4}{14n+3}$ is irreducible. Q.E.D.

**Detailed informal solution.** Denoting the greatest common divisor of $a, b$ as $(a, b)$, we use the Euclidean algorithm: $(21n + 4, 14n + 3)$.
First, we calculate the remainder when $21n + 4$ is divided by $14n + 3$:
$21n + 4 = (14n + 3) \cdot 1 + (7n + 1)$, so $(21n + 4, 14n + 3) = (14n + 3, 7n + 1)$.
Next, we calculate the remainder when $14n+3$ is divided by $7n+1$: $14n+3 = (7n+1) \cdot 2+1$, so $(14n+3, 7n+1) = (7n+1, 1)$.
Finally, since the greatest common divisor of any number and 1 is 1, we have $(7n + 1, 1) = 1$.
Therefore, $(21n + 4, 14n + 3) = 1$.
It follows that $\frac{21n+4}{14n+3}$ is irreducible. Q.E.D.

**Formal solution.**

```
theorem:
  fixes n :: nat
  shows "gcd (21*n + 4) (14*n + 3) = 1"
proof -
  (* Denoting the greatest common divisor of a, b as (a,b), we use the Euclidean algorithm: *)
    (* (21n+4, 14n+3). *)
    (* First, we calculate the remainder when 21n+4 is divided by 14n+3: *)
  have step1: "21*n + 4 = (14*n + 3) * 1 + (7*n + 1)"
    by simp
  (* 21n+4 = (14n+3) \cdot 1 + (7n+1), so (21n+4, 14n+3) = (14n+3, 7n+1). *)
  have step2: "gcd (21*n + 4) (14*n + 3) = gcd (14*n + 3) (7*n + 1)"
    using gcd_non_0_nat[of "14*n + 3" "7*n + 1"] step1
    by (metis gcd.commute gcd_add2 nat_mult_1_right)
  (* Next, we calculate the remainder when 14n+3 is divided by 7n+1: *)
  have step3: "14*n + 3 = (7*n + 1) * 2 + 1"
    by simp
  (* 14n+3 = (7n+1) \cdot 2 + 1, so (14n+3, 7n+1) = (7n+1, 1). *)
  have step4: "gcd (14*n + 3) (7*n + 1) = gcd (7*n + 1) 1"
    using gcd_non_0_nat[of "7*n + 1" 1] step3
    by (metis gcd.bottom_right_bottom gcd.commute gcd_add_mult mult.commute nat_1_add_1)
  (* Finally, since the greatest common divisor of any number and 1 is 1, we have (7n+1, 1) = 1. *)
  have step5: "gcd (7*n + 1) 1 = 1"
    by simp
  (* Therefore, (21n+4, 14n+3) = 1. *)
  have "gcd (21*n + 4) (14*n + 3) = 1"
    using step2 step4 step5
    by simp
  (* It follows that \frac{21n+4}{14n+3} is irreducible. The result follows. *)
  then show ?thesis
    by simp
qed
```

*Figure 3.* First example of a number theory problem solved by SPADER.

**Problem.** What is the units digit of the sum of the squares of the first nine positive integers? Show that it is 5.

**Original informal solution.** We can reduce the amount of work we have to do in this problem by realizing that the units digit of the sum of the squares is the units digit of the sum of the units digits of the squares. In other words, the units digit of $1^2 + 2^2 + \ldots + 9^2$ is the units digit of $1 + 4 + 9 + 6 + 5 + 6 + 9 + 4 + 1 = 45$, which is 5.

**Detailed informal solution.** We can reduce the amount of work we have to do in this problem by realizing that the units digit of the sum of the squares is the units digit of the sum of the units digits of the squares.
To find the units digit of $1^2 + 2^2 + \ldots + 9^2$, we first compute the units digit of each square:
- The units digit of $1^2$ is 1.
- The units digit of $2^2$ is 4.
- The units digit of $3^2$ is 9.
- The units digit of $4^2$ is 6.
- The units digit of $5^2$ is 5.
- The units digit of $6^2$ is 6.
- The units digit of $7^2$ is 9.
- The units digit of $8^2$ is 4.
- The units digit of $9^2$ is 1.
Summing these units digits, we get $1 + 4 + 9 + 6 + 5 + 6 + 9 + 4 + 1 = 45$.
Thus, the units digit of the sum $1^2 + 2^2 + \ldots + 9^2$ is the same as the units digit of $45$, which is 5.

**Formal solution.**

```
theorem mathd_numbertheory_3 :
  "(\<Sum> x < 10. ((x + 1)^2)) mod 10 = (5::nat)"
theorem mathd_numbertheory_3 :
  "(\<Sum> x < 10. ((x + 1)^2)) mod 10 = (5::nat)"
proof -
  (* We can reduce the amount of work we have to do in this problem by realizing that the units digit of the sum of the squares is the units digit of
      the sum of the units digits of the squares. *)
  (* To find the units digit of 1^2+2^2+\ldots+9^2, we first compute the units digit of each square: *)
  (* The units digit of 1^2 is 1. *)
  have c1: "(1^2 :: nat) mod 10 = 1"
    by simp
  (* The units digit of 2^2 is 4. *)
  have c2: "(2^2 :: nat) mod 10 = 4"
    by simp
  (* The units digit of 3^2 is 9. *)
  have c3: "(3^2 :: nat) mod 10 = 9"
    by simp
  (* The units digit of 4^2 is 6. *)
  have c4: "(4^2 :: nat) mod 10 = 6"
    by simp
  (* The units digit of 5^2 is 5. *)
  have c5: "(5^2 :: nat) mod 10 = 5"
    by simp
  (* The units digit of 6^2 is 6. *)
  have c6: "(6^2 :: nat) mod 10 = 6"
    by simp
  (* The units digit of 7^2 is 9. *)
  have c7: "(7^2 :: nat) mod 10 = 9"
    by simp
  (* The units digit of 8^2 is 4. *)
  have c8: "(8^2 :: nat) mod 10 = 4"
    by simp
  (* The units digit of 9^2 is 1. *)
  have c9: "(9^2 :: nat) mod 10 = 1"
    by simp
  (* Summing these units digits, we get 1 + 4 + 9 + 6 + 5 + 6 + 9 + 4 + 1 = 45. *)
  have "1 + 4 + 9 + 6 + 5 + 6 + 9 + 4 + 1 = (45::nat)"
    by simp
  (* Thus, the units digit of the sum 1^2 + 2^2 + \ldots + 9^2 is the same as the units digit of 45, which is 5. *)
  hence "(1 + 4 + 9 + 6 + 5 + 6 + 9 + 4 + 1) mod 10 = (45 mod 10)"
    by simp
  also have "... = (5::nat)"
    by simp
  (* The result follows. *)
  then show ?thesis
    by eval
qed
```

*Figure 4.* Second example of a number theory problem solved by SPADER.

**Problem.** If $a, b,$ and $c$ are positive real numbers such that $a(b + c) = 152$, $b(c + a) = 162$, and $c(a + b) = 170$, then $abc$ is
(A) 672     (B) 688     (C) 704     (D) 720     (E) 750
Show that it is 720.

**Original informal solution.** Adding up the three equations gives $2(ab + bc + ca) = 152 + 162 + 170 = 484 \implies ab + bc + ca = 242$.
Subtracting each of the above equations from this yields, respectively, $bc = 90$, $ca = 80$, $ab = 72$.
Taking their product, $ab \cdot bc \cdot ca = a^2 b^2 c^2 = 90 \cdot 80 \cdot 72 = 720^2 \implies abc = 720 \Rightarrow$ (D).

**Detailed informal solution.** Adding up the three equations gives $2(ab + bc + ca) = 152 + 162 + 170 = 484 \implies ab + bc + ca = 242$.
Subtracting each of the above equations from this yields, respectively, $ab + bc + ca - a(b + c) = 242 - 152 \implies bc = 90$, $ab + bc + ca - b(c + a) = 242 - 162 \implies ca = 80$, and $ab + bc + ca - c(a + b) = 242 - 170 \implies ab = 72$.
Taking their product, $(ab) \cdot (bc) \cdot (ca) = (abc)^2 = 72 \cdot 90 \cdot 80$.
Calculating the product, $72 \cdot 90 = 6480$ and $6480 \cdot 80 = 518400 \implies (abc)^2 = 518400 \implies abc = \sqrt{518400} = 720 \Rightarrow$ (D).
The result follows.

**Formal solution.**

```
theorem:
  fixes a b c::real
  assumes h0: "0 < a \<and> 0 < b \<and> 0 < c"
    and h1: "a * (b + c) = 152"
    and h2: "b * (c + a) = 162"
    and h3: "c * (a + b) = 170"
  shows "a * b * c = 720"
proof -
  (* Adding up the three equations gives 2(ab + bc + ca) = 152 + 162 + 170 = 484 *)
  have c0: "2 * (a * b + b * c + c * a) = 152 + 162 + 170"
    using h1 h2 h3
    by (simp add: algebra_simps)
  (* \Longrightarrow ab + bc + ca = 242 *)
  hence c1: "a * b + b * c + c * a = 242"
    by simp
  (* Subtracting each of the above equations from this yields, respectively, *)
  (* ab + bc + ca - a(b+c) = 242 - 152 \Longrightarrow bc = 90 *)
  have c2: "a * b + b * c + c * a - a * (b + c) = 242 - 152"
    using c1 h1
    by simp
  hence c3: "b * c = 90"
    by (auto simp: field_simps algebra_simps)
  (* ab + bc + ca - b(c+a) = 242 - 162 \Longrightarrow ca = 80 *)
  have c4: "a * b + b * c + c * a - b * (c + a) = 242 - 162"
    using c1 h2
    by simp
  hence c5: "c * a = 80"
    by (auto simp: field_simps algebra_simps)
  (* ab + bc + ca - c(a+b) = 242 - 170 \Longrightarrow ab = 72 *)
  have c6: "a * b + b * c + c * a - c * (a + b) = 242 - 170"
    using c1 h3
    by simp
  hence c7: "a * b = 72"
    by (auto simp: field_simps algebra_simps)
  (* Taking their product, (ab) \cdot (bc) \cdot (ca) = (abc)^2 = 72 \cdot 90 \cdot 80. *)
  have c8: "(a * b) * (b * c) * (c * a) = (a * b * c) ^ 2"
    by (simp add: power2_eq_square)
  hence c9: "(a * b * c) ^ 2 = 72 * 90 * 80"
    using c3 c5 c7
    by simp
  (* Calculating the product, 72 \cdot 90 = 6480 and 6480 \cdot 80 = 518400 \Longrightarrow (abc)^2 = 518400 *)
  hence c10: "(a * b * c) ^ 2 = 518400"
    by (simp add: mult.commute mult.left_commute)
  (* \Longrightarrow abc = \sqrt{518400} = 720 *)
  hence c11: "a * b * c = sqrt 518400"
    by (simp add: h0 less_eq_real_def real_sqrt_unique)
  hence "a * b * c = 720"
    by simp
  (* The result follows. *)
  then show ?thesis
    by simp
qed
```

*Figure 5.* First of an algebra problem solved by SPADER.

**Problem.** Expand the product $(x + 1)^2 \cdot x$. Show that it is $x^3 + 2x^2 + x$.

**Original informal solution.** We have $(x + 1)^2 = (x + 1)(x + 1) = x(x) + 1(x) + 1(x) + 1 = x^2 + 2x + 1$. Multiplying this by $x$ gives $x^3 + 2x^2 + x$.

**Detailed informal solution.** We have $(x+1)^2 = (x+1)(x+1) = x \cdot x + x \cdot 1 + 1 \cdot x + 1 \cdot 1 = x^2 + x + x + 1 = x^2 + 2x + 1$. Multiplying this by $x$ gives $x \cdot (x^2 + 2x + 1) = x \cdot x^2 + x \cdot 2x + x \cdot 1 = x^3 + 2x^2 + x$. The result follows.

**Formal solution.**

```
theorem:
  fixes x ::real
  shows "(x + 1)^2 * x = x^3 + 2 * x^2 + x"
proof -
  (* We have (x+1)^2 = (x+1)(x+1) *)
  have c0: "(x + 1)^2 = (x + 1) * (x + 1)"
    by (simp add: power2_eq_square)
  (* = x \cdot x + x \cdot 1 + 1 \cdot x + 1 \cdot 1 *)
  also have "... = x * x + x * 1 + 1 * x + 1 * 1"
    by (simp add: algebra_simps)
  (* = x^2 + x + x + 1 *)
  also have "... = x^2 + x + x + 1"
    by (simp add: power2_eq_square)
  (* = x^2 + 2x + 1 *)
  also have "... = x^2 + 2 * x + 1"
    by (simp add: algebra_simps)
  finally have c1: "(x + 1)^2 = x^2 + 2 * x + 1"
    .
  (* Multiplying this by x gives x \cdot (x^2 + 2x + 1) *)
  have c2: "(x^2 + 2 * x + 1) * x = x * (x^2 + 2 * x + 1)"
    by (simp add: algebra_simps)
  (* = x \cdot x^2 + x \cdot 2x + x \cdot 1 *)
  also have "... = x * x^2 + x * (2 * x) + x * 1"
    by (simp add: algebra_simps)
  (* = x^3 + 2x^2 + x *)
  also have "... = x^3 + 2 * x^2 + x"
    by (smt (verit, ccfv_SIG) One_nat_def Suc_1 \<open>x * x + x * 1 + 1 * x + 1 * 1 = x\<^sup>2 + x
        + x + 1\<close> mult.commute numeral_3_eq_3 power.simps(2) ring_class.ring_distribs(2))
  finally have "(x^2 + 2 * x + 1) * x = x^3 + 2 * x^2 + x"
    .
  (* The result follows. *)
  then show ?thesis
    using c1 c2
    by simp
qed
```

*Figure 6.* Second example of an algebra problem solved by SPADER.

