# OpenReview forum: "More Details, Please: Improving Autoformalization with More Detailed Proofs"
_ICML.cc/2024/Workshop/AI4MATH — ICML 2024 Workshop AI4MATH Poster_

### Official Review · Reviewer_GidZ · 2024-06-09

**Rating:** 6
**Confidence:** 4

**Summary:**

This paper describes a methodology for proof autoformalization using an iterative sketching and informal proof repair procedure. Given an informal theorem statement and proof, and a formal statement in Isabelle, they generate a formal sketch in the style of DSP. Automated reasoning tools attempt to fill in the sketch, and locations where failures occur are mapped to the informal proofs. The informal proofs are then rewritten with more detail and passed for another round of the process. They show marginal improvement over the (draft) sketch prove baseline, with no evidence as to the necessity of the iterativeness of the procedure.

**Questions:**

1. Could you include a baseline comparing Decomposing the Enigma vs. SPADeR? It seems that an additional informal proof refinement may be unnecessary, as you can begin with a refinement at the start of the process.
2. Can you clarify this sentence in the related works section: "None of these methods, however, can make use of informal mathematical data, which is significantly more abundant than formal data." Is the intended purpose to claim that SPADeR is capable of using informal data? If so, the same is true of all methods listed in the autoformalization section, yet I think an important distinction should be made with whether these methods are better understood as autoformalizers rather than theorem-provers due to potential test-set contamination.
3. The authors mention in the appendix that Figure 3 corresponds to a theorem which could not be proven in the DSP paper using a human proof, but can be with an automatically generated proof. Can the authors clarify whether they are using human-informal proofs, as there does not seem to be an initial drafting phase, and whether *their* implementation of the Sketch and Prove baseline is not capable of proving this theorem, but SPADeR is?

**Reasons To Accept:**

1. The paper describes a natural modification to the DSP pipeline and provides experimental results with that modification, even if the results are not generally positive. Such information can be useful for the research community.

**Reasons To Reject:**

1. The results indicate marginal-at-best improvement over the sketching baseline. The iterative nature of the approach is not demonstrated to be useful. They mention Lyra in the related works, which also performs an iterative procedure based on amending formal proofs given erroneous data, which does yield some positive results with multiple passes. However, the alteration method here does not seem to help with multiple passes.
2. The idea of adding further detail to the informal proof is not new, and is done in Decomposing the Enigma (https://arxiv.org/abs/2305.16366) where the informal proof is enhanced at the start, rather than only making modifications after the automated reasoning tools fail.

---

### Official Review · Reviewer_knhC · 2024-06-10

**Rating:** 7
**Confidence:** 5

**Summary:**

The paper introduces SPADER (Sketch, Prove, Add Details & Repeat), a method to enhance autoformalization of mathematical proofs using Large Language Models (LLMs). The approach aims to bridge the gap between informal and formal proofs by using LLMs to infer and incorporate implicit details from informal proofs, thereby increasing the success rate of formal verification. The method was tested on the miniF2F dataset, showing an improvement in the percentage of successfully formalized problems from 34.8% to 38.1%. The paper contributes by proposing a detailed-oriented approach to autoformalization and demonstrating its effectiveness in improving the accuracy of language model-based theorem provers.

**Questions:**

Your results indicate that multiple detailing passes (M = 2) do not significantly improve performance over a single pass (M = 1). Could you discuss why this might be the case and whether there are scenarios where multiple passes could be more beneficial?

**Reasons To Accept:**

The paper presents a novel and significant contribution to the field of automated theorem proving by addressing a critical gap in the transition from informal to formal proofs. The experimental results demonstrate a clear improvement in success rates, and the methodology is well-documented and transparent. While there are areas for improvement, such as broader evaluation, detailed model training information, and scalability analysis, these are not fundamental flaws that undermine the validity or significance of the work. The suggestions provided can be addressed in revisions. The proposed method, SPADER, has the potential to significantly advance the state of the art in autoformalization and automated theorem proving, making it a valuable addition to the research community.

**Reasons To Reject:**

n/a

---

### Official Review · Reviewer_2aFf · 2024-06-12

**Rating:** 6
**Confidence:** 4

**Summary:**

This paper proposes a novel framework for autoformalization on top of DSP. The primary motivation comes from the misalignment between the reasoning steps in informal proofs and formal sketches, where one step in natural language might correspond to multiple formal codes in Isabelle. Therefore, the paper suggests adding an additional pipeline: whenever a specific conjecture in the proof sketch fails to pass Isabelle's proof checking, the LLM will be prompted to expand the informal proof of this sketch and extend the formalization of this step. The experimental results show that the proposed method is effective, improving the pass rate compared to the ablation counterpart.

**Questions:**

Please justify the question I raised in the `Reasons To Reject`.

**Reasons To Accept:**

- The proposed framework is interesting and novel, with good motivation, and it addresses an important problem in autoformalization.
- The experiments demonstrate its effectiveness on the miniF2F dataset.

**Reasons To Reject:**

The experimental result using GPT-4o is relatively low. In my understanding, the Sketch and Prove baseline is the same as DSP with a human-written informal proof. However, DSP achieves a 39.3% pass rate on the miniF2F test with the Codex model, whereas the Sketch and Prove baseline with GPT-4o achieves only a 34.8% pass rate. This is unlikely to happen, as GPT-4o has significant improvements compared to Codex and other models. One experimental result on GPT-4 (not GPT-4o) with DSP can be seen in the [Lyra](https://arxiv.org/abs/2309.15806) or [LEGO-Prover](https://openreview.net/forum?id=3f5PALef5B) papers, where GPT-4 achieves a 43.0% pass rate with model-generated proof (the result on human-written proof should be comparable). I don't understand why the pass rate for GPT-4o is so low with Sketch and Prove and SPADER.

---

### Official Review · Reviewer_sWwG · 2024-06-13

**Rating:** 5
**Confidence:** 4

**Summary:**

The paper proposes a novel method for autoformalization, using language models to infer and explicitly incorporate implicit details from informal proofs. The percentage of successfully formalized problems increases from 34.8% to 38.1%.

**Questions:**

The success rate in [DRAFT, SKETCH, AND PROVE](https://arxiv.org/pdf/2210.12283) paper with the "Human informal proof" setting is 39.3%, why does your paper have only 34.8%? Is GPT-4o weaker than Codex used in DSP? Can you reimplement the score of DSP?

**Reasons To Accept:**

1. The proposed method is easy to implement and has significant improvements compared to the baseline.
2. The writing is clear and easy to follow.

**Reasons To Reject:**

More models should be tested.

---

### Meta-Review · Area_Chair_fuiZ · 2024-06-13

**Recommendation:** Accept (Poster)
**Confidence:** 4

**Metareview:**

The paper proposes a novel method for autoformalization based on DSP, using an iterative manner to fix informal proof with failure details. The proposed method marginal improvement over the DSP baseline. The paper describes a natural modification to the DSP pipeline and the not generally positive results can be useful for the research community. To improve the paper, the author needs to clarify the GPT-4o results, Figure 3, and the statement about informal data usage in the related works section.

---

### Decision · Program_Chairs · 2024-06-13

Accept (Poster)